# The Therapeutic Effectiveness of Full Spectrum Hemp Oil Using a Chronic Neuropathic Pain Model

**DOI:** 10.3390/life10050069

**Published:** 2020-05-18

**Authors:** Jacob M. Vigil, Marena A. Montera, Nathan S. Pentkowski, Jegason P. Diviant, Joaquin Orozco, Anthony L. Ortiz, Lawrence J. Rael, Karin N. Westlund

**Affiliations:** 1Department of Psychology, University of New Mexico, Albuquerque, NM 87131, USA; vigilj@unm.edu (J.M.V.); pentkowski@unm.edu (N.S.P.); diviantj@unm.edu (J.P.D.); orozcoj@unm.edu (J.O.); 2Department of Anesthesiology, University of New Mexico Health Sciences Center, Albuquerque, NM 87131, USA; monteram@salud.unm.edu; 3Organic-Energetic Solutions, Albuquerque, NM 87108, USA; aortiz@Kurplemag.com (A.L.O.); coloradohemppharm@yahoo.com (L.J.R.)

**Keywords:** endocannabinoids, cannabidiol, pain, allodynia, *Cannabis*, hemp

## Abstract

Background: Few models exist that can control for placebo and expectancy effects commonly observed in clinical trials measuring ‘*Cannabis*’ pharmacodynamics. We used the Foramen Rotundum Inflammatory Constriction Trigeminal Infraorbital Nerve injury (FRICT-ION) model to measure the effect of “full-spectrum” whole plant extracted hemp oil on chronic neuropathic pain sensitivity in mice. Methods: Male BALBc mice were submitted to the FRICT-ION chronic neuropathic pain model with oral insertion through an incision in the buccal/cheek crease of 3 mm of chromic gut suture (4-0). The suture, wedged along the V2 trigeminal nerve branch, creates a continuous irritation that develops into secondary mechanical hypersensitivity on the snout. Von Frey filament stimuli on the mouse whisker pad was used to assess the mechanical pain threshold from 0–6 h following dosing among animals (n = 6) exposed to 5 μL of whole plant extracted hemp oil combined with a peanut butter vehicle (0.138 mg/kg), the vehicle alone (n = 3) 7 weeks post-surgery, or a naïve control condition (n = 3). Results: Mechanical allodynia was alleviated within 1 h (d = 2.50, *p* < 0.001) with a peak reversal effect at 4 h (d = 7.21, *p* < 0.001) and remained significant throughout the 6 h observation window. There was no threshold change on contralateral whisker pad after hemp oil administration, demonstrating the localization of anesthetic response to affected areas. Conclusion: Future research should focus on how whole plant extracted hemp oil affects multi-sensory and cognitive-attentional systems that process pain.

## 1. Introduction

The enactment of the Hemp Farming Act, effectively beginning in 2019, was a monumental milestone in the history of *Cannabis* prohibition in the United States (U.S.), because it enabled the legal consumption, commercial production, and market trade of any type of product made from certain variants of the *Cannabis* plant. Only differing from their federally illegal counterparts, arbitrarily defined as *Cannabis* plants containing over 0.3% tetrahydrocannabinol (THC) potency levels, the legal variety of the *Cannabis* plant—conventionally referred to as ‘hemp’—still contains hundreds of additional phytochemicals, including cannabinoids (e.g., cannabidiols, CBDs), terpenes, terpenoids, and flavonoids that may offer potent therapeutics, both individually and synergistically [1,2,3,4]. Despite widespread *Cannabis* usage in the U.S., with estimated annual revenues now in the tens of billions of dollars, current patients and providers still have little scientific evidence on the likely effectiveness of common and commercially available cannabis-based products for pharmaceutical application. This is because the federal government has largely limited clinical investigations to plant-inspired isolates, concocted formulants, or synthetic analogues not representative of the whole, natural *Cannabis* plant-based products most widely used by millions of people in the U.S. [5,6]. Another frequent and unavoidable limitation of extant human trials measuring *Cannabis*’ pharmacodynamics is that they cannot control for placebo and expectancy effects, or visceral, perceptual, and/or cognitive reactions to enrollment in a cannabis-themed experiment, with several studies observing *Cannabis*-related experiences reported by both active agent and placebo group participants [7,8].

While animal models can control for expectancy effects, few paradigms have created a persistent state of chronic pain, with the majority of conventional pain models resulting in a recoverable, and hence qualitatively different forms of pain than one that is ‘chronic’ in nature, and hence often and uniquely tethered with comorbid affective disturbances (e.g., depression) [9,10]. One well-established and reliable chronic pain model, the Foramen Rotundum Inflammatory Constriction Trigeminal Infraorbital Nerve injury (FRICT-ION) model involves an insertion of 3 mm of chromic gut suture (4-0) along the maxillary branch as it passes into the foramen rotundum through a tiny scalpel incision in the buccal/cheek crease [11]. Mechanical hypersensitivity reliably develops on the snout persisting through >100 days, likely due to consistent inflammatory response caused in part by movement of the nerve during chewing. The extended 3–10 week timeframe for study allowed by the FRICT-ION model is reportedly equivalent to 5–8 years of chronic pain in clinical patients [12,13]. 

Studies examining the neuropharmacology of neuropathic pain have implicated opioid (e.g., MOP/DOP) [14,15,16], serotonin (e.g., 5-HT_7_) [17,18], dopamine (e.g., D_2_) [19,20] and glutamate (e.g., GluN2B) [21,22,23] receptor systems as potential therapeutic targets; however, no studies to date have examined the effects of whole plant-extracted hemp oil on chronic pain. Therefore, in the present study, we investigated the analgesic effects of “full-spectrum” whole plant oil extracted from the hemp plant, using ethanol and evaporation-based procedures commonly used in the *Cannabis* industry, on mechanical neuropathic chronic pain sensitivity in mice. By creating a continuous state of irritation in the infraorbital nerve, the FRICT-ION mouse model of chronic orofacial neuropathic pain can initiate mechanical allodynia in the mouse whisker pad for pharmaceutical investigation. We use a standard von Frey test for mechanical hypersensitivity at 7 weeks post-surgery to measure the effects of orally administered hemp oil over a 6 h observation window, in comparison to vehicle only and naïve control mice, to estimate the general efficacy of commonly used hemp-based products for therapeutic application.

## 2. Materials and Methods

### 2.1. Experimental Plan 

The general experimental plan is shown in Figure 1. 

Model induction began 1 week after acclimatization of the male mice. The animals were tested weekly for mechanical sensitivity with von Frey fibers applied to the whisker pad. Hypersensitivity was induced within the first week and continued indefinitely. After the week 7 testing, mice with FRICT-ION nerve compression neuropathic pain were given either hemp oil mixed in peanut butter (0.138 mg/kg) or peanut butter alone. Naïve control mice received weekly behavioral testing.

### 2.2. Animals 

Subjects were 12 male BALBc mice (20–30 g, Envigo) housed in the animal housing facility of the University of New Mexico Health Science Campus. Mice were housed in groups of 3 per cage, with food and water available ad libitum, in a temperature-controlled room (23 °C) under a reversed 12 h/12 h light/dark cycle (lights on at 7:00 PM). The mice were transported to the experimental room in their home cages and left undisturbed for 1 h prior to testing. All efforts were made to minimize animal stress and suffering, and to reduce the number of mice used. The experiment received formal approval from the Committee on Animal Research and Ethics, Health Sciences Center, University of New Mexico (IACUC #17-200613). The experiment reported in this article was performed in accordance with the recommendations of the United States National Institutes of Health Guide for the Care and Use of Laboratory Animals and the guidelines of the Committee for Research and Ethical Issues of the International Association for the Study of Pain [24].

### 2.3. Surgical Procedures for Induction of the Model

The surgical model induction has been described in detail previously [11]. Briefly, mice were anesthetized with isoflurane (2%) and then a small surgical incision was made orally between the cheek and maxillary bone. A chromic gut suture (4.0, 3 mm) was inserted into the tight space between the infraorbital (ION) trigeminal nerve branch and the bony foramen rotundum as the nerve enters the skull. The anesthesia and surgery took less than 5 min. All surgeries were performed in sterile conditions under a surgical microscope. 

Whisker pad hypersensitivity developed reliably and persisted for at least 3 months [11,25]. In sham operations reported in a previous full description of the procedure, the ION was exposed but not compressed [11]. All animals underwent weekly behavioral testing to confirm mechanical threshold level and sham animals responded like naïve animals after recovery from the sham surgery (Figure 1 and Figure 2A). Estimates are that in week 8, mice had experienced pain equivalent to several human years and can be considered chronic [12,13]. Group comparisons included naïve and neuropathic pain groups, with and without hemp oil. All studies were blinded.

### 2.4. Assessment of Mechanical Allodynia

Post-surgery mechanical sensitivity of the whisker pad was determined in the infraorbital nerve receptive field with a series of 8 von Frey fiber filaments (0.008 g (1.65); 0.02 g (2.36); 0.07 g (2.83); 0.16 g (3.22); 0.4 g (3.61); 1.0 g (4.08); 2.0 g (4.31); 6.0 g (4.74); Stoelting, Wood Dale, IL, USA) by the modified up-down method [26]. Mice were handled several times before experiments and were trained in the procedure for one week prior to the actual test. The experimenter held the mouse with a cupped hand until the animal was calm. Mice moved freely in the holder’s hands with its head exposed. During testing, the experimenter slightly restrained the mouse in their hand while accurately applying the von Frey filament onto the center of the mouse whisker pads, both ipsilateral and contralateral to the surgery site. For the consistency of results, each filament was applied five times at intervals of a few seconds. If head withdrawal was observed at least three times after probing with a filament, the mouse was considered responsive to that filament according to the up-down method [26]. For this approach, whenever a positive response to the mechanical stimulus occurred, the next weaker von Frey filament was applied. If no positive response was evoked, the next stronger filament was applied. Testing proceeded in this manner until four fibers applied after the first one successfully caused positive responses. This allowed estimation of the 50% mechanical withdrawal threshold (in grams force) using a curve-fitting algorithm. Baseline assessment of nociception was done in week 2 after acclimatization. After surgical induction of the FRICT-ION model, testing for hypersensitivity was done once weekly through to week 12.

### 2.5. Hemp Oil Dosing

Prior to testing the effects of hemp oil on the behavioral changes, mechanical allodynia was confirmed in the mice in weeks 1–6 after induction of the nerve trauma and re-tested in week 7, prior to the experiment (Figure 2A). The mechanical pain threshold was measured hourly (0–6 h) following dosing among animals exposed to 5 μL of hemp oil dissolved in peanut butter as the vehicle (0.138 mg/kg; n = 6), the peanut butter vehicle alone (n = 3), or a naïve control condition (n = 3) (Figure 2B). The naïve mice underwent all procedures and tests except the surgery. The animals immediately ate the peanut butter with or without the hemp oil. The mechanical threshold on the whisker pad on the side of the inflamed nerve and on the contralateral side was assessed hourly after consumption for 6 h.

### 2.6. Hemp Cultivation and Oil Extraction 

The hemp oil was cultivated by Organic-Energetic Solutions LLC (Albuquerque, NM, USA) and commercially available under the “LyFeBaak” label. The oil is derived from “Cherry Blossom” hemp plants that have been cultivated using proprietary methods that hypermineralize the plants throughout the vegetative and flowering phases of cultivation. The cannabinoid and terpene analytics of the Cherry Blossom plants used to produce the hemp oil are shown in Table 1 and Table 2, respectively. Mature *Cannabis* hemp buds were used in an alcohol bath extraction procedure using ethanol as a solvent. The procedure stripped the cannabinoids, terpenes, chlorophyll, fats, lipids and wax compounds from plant material by suspending those compounds in an alcohol solution form. The solution was then separated from the plant material using a strainer and the alcohol solution was further filtered until it was free of particulates. The alcohol was evaporated out of the solution, leaving a highly concentrated form of the botanical components harvested from the *Cannabis* plants, which was then combined with medium-chain triglyceride (MCT) oil in its final (retail) form. In the current study, 5 μL of hemp oil were combined with peanut butter (0.138 mg/kg) for the active agent group.

### 2.7. Statistical Analysis

The Prism 4 statistical program was used for data analysis (Graph Pad Software, Inc., La Jolla, CA, USA). All data were expressed as mean ± SD. The hourly changes in pain threshold following drug consumption were analyzed by one-way ANOVA followed by Tukey’s multiple comparison post hoc tests. A *p* ≤ 0.05 was considered significant. 

## 3. Results

The naïve group of mice responded throughout the testing period only to von Frey fibers which apply 6.000 g force (blue line, Figure 2) (6.000 ± 0.000). This was considered the baseline response. The development and continuation of whisker pad hypersensitivity in the untreated group is shown in Figure 2A,B as the red line, indicating a response threshold to a von Frey fiber with minimal grams force (0.008 g) (0.067 ± 0.029). A repeated measures ANOVA showed a significant group–time interaction, F (2,9) = 17.65, *p* < 0.001). The treated group had the same response as the untreated group initially prior to the administration of the hemp oil, but the response reversed toward baseline stably between 2–4 h (Figure 2B green line). The response thresholds showed a bell-shaped dose-response curve in the treated group, but did not change over the 6 h observation period for the naïve and untreated groups (*p* > 0.10 in each case). 

The results indicate the efficacy of the drug treatment in the chronic trigeminal neuropathic pain model. As shown in Figure 2B, mechanical withdrawal threshold assessment indicated relief from hypersensitivity following hemp oil treatment at 1 (d = 2.50, *p* < 0.001), 2 (d = 3.44, *p* < 0.001), and 6 h (d = 2.20, *p* < 0.001), with a peak reversal of allodynia at 4 h (d = 7.21, *p* < 0.001). No adverse events were observed.

## 4. Discussion

There is an urgent need for novel, alternative, opioid-sparing pain medications [27]. To date, a growing body of evidence supports the use of *Cannabis*-based medicines for modulating analgesic, anti-inflammatory and anxiolytic effects [28,29,30,31,32,33,34], with chronic pain being the most widely cited health condition for medical *Cannabis* usage [5,35,36,37]. The current findings are consistent with clinical research among chronic pain patients [38] and large population-based studies measuring the real-time effects of *Cannabis* consumption on momentary pain intensity [39,40], which show that *Cannabis* is an effective mid-level analgesic. However, the current study is unique in measuring the effect of full spectrum hemp-extracted oil with negligible THC levels. Using an established and reliable chronic pain mouse model, we circumvented the unavoidable confound of human reactivity present in all clinical trials, and showed that 5 μL of legal hemp oil is a potent analgesic, reducing mechanical pain sensitivity over tenfold. The FRICT-ION model is an improvement over other trigeminal nerve injury models [41,42]. The model is minimally invasive, is induced in 5–10 min, reducing anesthetic exposure, persists indefinitely, and can easily be blinded compared to other surgical models where the animals have external sutures and shaved fur [41,42]. The current study builds on studies measuring the therapeutics of isolated or formulated cannabinoids, and most popularly, CBD [43], by showing that common and commercially available full spectrum oil extracted from the hemp plant, using techniques that are generally replicable and accessible to the layperson, may be an effective treatment for chronic pain caused by mechanical pressure. The study addresses the therapeutic value of hemp using an in vivo pain model that mimics injury to the trigeminal nerve and causes pain behaviors analogous to human chronic neuropathic pain [44]. Pain arising from insult to the maxillary division of the trigeminal nerve has been lauded as a good predictor of efficacy for pain therapeutics that are later guided through FDA approvals [45]. However, given that chronic pain remains a significant clinical challenge with a conventional treatment response rate of only 11% [46,47], the current findings show promising support for the effectiveness of a recently legalized (in the U.S.) alternative ethnopharmaceutical option for the treatment of some forms of chronic pain. 

One of the most widely studied constituents of the *Cannabis* plant, CBD, may attenuate pain through several mechanisms of action. Although not fully characterized, the pharmacodynamic profile of CBD is diverse, producing agonistic and antagonistic effects at multiple cannabinoid and non-cannabinoid receptor sites. For instance, despite somewhat low affinity for the orthosteric binding sites, at low nanomolar doses, CBD is a non-competitive negative allosteric modulator at CB_1_ receptors [48,49,50] and an inverse agonist at CB_2_ receptors [49,51,52]. Additional non-cannabinoid mechanisms of CBD action include: (1) agonism at serotonin-1A (5-HT_1A_) [53,54] and transient receptor potential cation (e.g., calcium) channels subfamily V member types 1 and 2 (TRPV1-2) receptors [55,56,57], (2) positive allosteric modulation at gamma amino butyric acid (GABA_A_) [58] and glycine receptors [59], (3) antagonism of voltage-gated Cav2.2 calcium channels [57] and sigma-1 receptors [60], and (4) indirect endocannabinoid agonism by increasing levels of N-arachidonoylethanolamine (AEA) via blocking reuptake and inhibiting breakdown by FAAH [56].

Based on previous reports, the analgesic effects of CBD detected in the present study likely involve mixed effects at various receptor types. For instance, CBD prevents paclitaxel-induced allodynia and carrageenan-induced thermal hyperalgesia, effects that are reversed by the 5-HT_1A_ antagonist WAY 100635 [61] and the TRPV1 antagonist capsazepine [62], respectively. Interestingly, the effects of CBD on thermal hyperalgesia do not involve activity at CB_1_ or CB_2_ receptors as selective CB antagonists do not prevent the antihyperalgesic effects of CBD. Similarly, repeated CBD administration for 7 days reduces spared nerve injury (SNI)-induced mechanical allodynia, which is prevented by capsazepine or WAY 100635 [61]. CBD antagonism of presynaptic Cav2.2 calcium channels [63] could also partially explain the present results, as selective CaV2.2 antagonists inhibit nociceptive signaling and reduce allodynia [64]. CBD also enhances morphine-evoked supraspinal antinociception in the tail-flick task, effects that depend on CBD antagonism of sigma-1 receptors [60]. CBD also reduces both CFA-induced inflammatory pain, surgical injury induced pain, and SNI-induced neuropathic pain, analgesic effects that depend on activation of the glycine alpha3 receptor [43,65]. Cannabidiolic acid (CBDA) and tetrahydrocannabinolic acid (THCA), the first and second most abundant cannabinoids (relatively) in the hemp plants used to produce the oil, have not been extensively investigated for their therapeutic value, due in part to their molecular instability, but have both been shown to influence nociceptive and nonnociceptive mechanoreceptors [66,67]. Collectively, these preclinical data indicate that CBD produces analgesic effects by altering signaling at multiple receptor systems and highlight the potential clinical utility of CBD for treating chronic pain.

In addition, terpenes such as β-myrcene, α-Pinene, and β-caryophyllene have been shown to exert anti-inflammatory and antinociceptive activities [68]. While CBD exerts prolonged immunosuppression and might be used in chronic inflammation, terpenes demonstrate a transient immunosuppression and relieve acute inflammation [69]. In our study, out of a total of 38 terpenoids tested, accounting for 0.65% of the hemp’s molecular weight, the eight most prominent terpenoids were: β-myrcene 0.256%, β-caryophyllene 0.121%, farnesene 0.080%, α-pinene 0.48%, guaiol 0.043%, α-bisabolol 0.040%, α-humulene 0.036%, and D-limonene 0.022%. Of these eight terpenoids, three have been shown to have either anti-inflammatory or antinociceptive properties, or both, and to be at a concentration of approximately 0.05% or greater, which is considered to be of pharmacological interest [1]. 

The monoterpene β-myrcene is one of the most prominent terpenes found in *Cannabis* and the most prominent terpene in the plants used to extract the hemp oil. It is often associated with sedating chemovars of cannabis, commonly referred to as “indicas.” β-myrcene has been shown to have high antinociceptive properties [70], and can be found in relatively high concentrations in plants such as lemongrass (*Cymbopogon citratus*), making up roughly 15% of dried product mass [71]. β-caryophyllene, in contrast, is one of the most common sesquiterpenoids found in *Cannabis* and the second most prominent terpenoid in the plants used to make the hemp oil. Various species from the genus *Copaifera* contain high concentrations of either β-caryophyllene or caryophyllene oxide (a metabolite of β-caryophyllene) [72], and have been shown to possess anti-inflammatory and analgesic properties [73]. While the exact mechanisms in which β-caryophyllene exerts its anti-inflammatory and antinociceptive properties are not fully understood, at least part of its mechanism of action is related to selective CB_2_ receptor agonism with no significant affinity for the CB_1_ receptor [74]. 

Finally, α-pinene is the most widely distributed terpenoid in nature and the fourth most prominent terpene in the Cherry Blossom plants used to make the hemp oil. In one study, α-pinene has been shown to significantly decrease lipopolysaccharide (LPS) induced production of nitric oxide, as well as interleukin-6 (IL-6) and tumor necrosis factor alpha (TNF-α), which are both pro-inflammatory cytokines. α-Pinene inhibited inducible nitric oxide synthase (iNOS) as well as cyclooxygenase-2 (COX-2) expression in LPS-stimulated macrophages [75]. In another study, an essential oil derived from the leaves of the black Chilean guava (*Ugni myricoides*) was analyzed and found to contain 52.1% α-pinene. The other constituents were 1,8-cineole (11.9%), α-humulene (4.6%), caryophyllene oxide + globulol (4.5%), humulene epoxide II (4.2%) and β-caryophyllene (2.9%) [76]. The essential oil was able to significantly prevent mechanical hypernociception induced by complete Freund’s adjuvant (CFA) in mice. Both the full spectrum essential oil and isolated α-pinene were able to abolish the mechanical sensitization induced by CFA or following the partial ligation of the sciatic nerve. The efficacy of the essential oil was described to be similar to that observed for indomethacin or gabapentin and thus of potential interest for the treatment of inflammatory and neuropathic pain [76]. 

The current study has several limitations, including the measurement of a single product derived by a single strain of hemp, rendering a comparison across different chemovars impossible. It is also not entirely clear if the results are consistent with combustion, intravenous and/or intramuscular dosing and future studies should examine the pharmacodynamics and pharmacokinetics of various types of consumption. Another general challenge for the health and scientific communities is the lack of standardization across natural *Cannabis*-based products due to the heterogeneous nature of ethnopharmaceuticals and individual differences in their pharmacodynamics, which can be affected by variability in metabolic, diurnal, and enzymatic functioning, as well as distinctions across individual’s microbiomes at the time of dosing. It is also unclear how the current results translate comparatively, given that human pain sensations can be affected by not only tissue damage (e.g., physical disease and injury) and nociception within the peripheral and central nervous systems (e.g., modulating afferent input), but also cognitive percepts and visceral sensations, moods, and situational and contextual factors such as social settings [77,78,79,80,81]. Finally, given the small sample and observational nature of the current research design, caution may be warranted when interpreting the generalizability of the findings and/or application to human conditions. Future research will benefit by comparing whole plant extracted hemp oil effects with dose-response curves of some of the main compounds present in the oil to verify the utility of full spectrum therapeutics.

## 5. Conclusions

The present study shows for the first time that common, commercially available, and easily reproducible full-spectrum hemp oil induces significant anti-allodynic effects with a bell-shaped pain sensitivity effect peeking between 2 and 4 h and lasting over 6 h. The study provides evidence that phytochemical extracts of the *Cannabis* plant, even with relatively low levels of THC, can significantly improve mechanical pressure pain in animals with established chronic neuropathic hypersensitivity. 

## Figures and Tables

**Figure 1 life-10-00069-f001:**
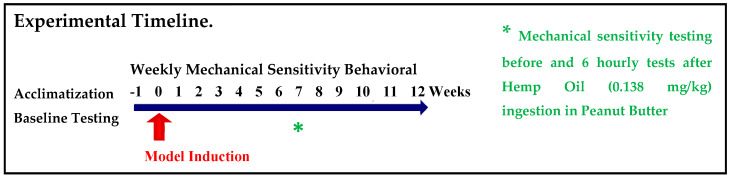
Experimental Timeline.

**Figure 2 life-10-00069-f002:**
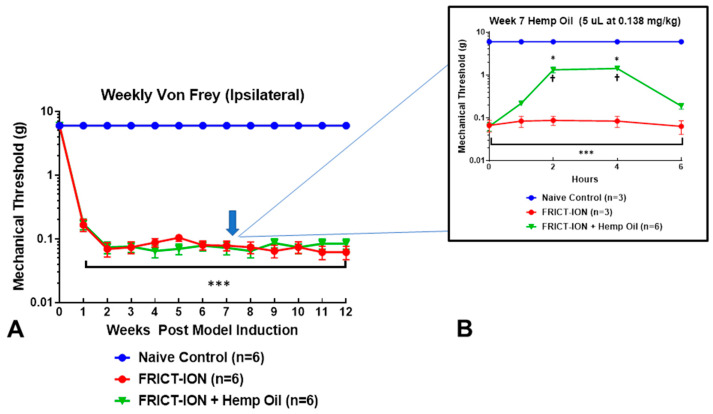
Mechanical hypersensitivity in a chronic trigeminal neuropathic pain model and effects of hemp oil. (**A**) Effect of trigeminal nerve injury. The reduction in mechanical threshold latency response to von Frey fiber mechanical stimulation of the whisker pad is statistically significant for 12 weeks after FRICT-ION trigeminal nerve compression injury. (**B**) Effect of hemp oil ingestion. The efficacy of hemp oil administered in peanut butter was tested in week 7 post nerve injury in male mice. Mechanical threshold was tested hourly on the ipsilateral whisker pad. The data are presented as a mean ± SEM. Two-way ANOVA with Dunnett’s multiple comparisons test. For the graphs, asterisks (*) indicate *p* < 0.05 comparing the FRICT-ION + Hemp to naïve control and (***) *p* < 0.001 comparing the FRICT-ION to naïve control. Dagger (†) indicates *p* < 0.05 comparing the FRICT-ION + Hemp to FRICT-ION.

**Table 1 life-10-00069-t001:** Cherry Blossom cannabinoid analytics.

Phytochemical	% Total Weight	mg/g
Cannabichromene (CBC)	0.120	1.200
Cannabigerolic Acid (CBGA)	0.281	2.810
Cannabigerol (CBG)	0.046	0.460
Tetrahydrocannabivarin (THCV)	0.022	0.220
*Delta-8-Tetrahydrocannabinol* (Δ8THC)	0.000	0.000
Cannabidivarin (CBDV)	0.000	0.000
Cannabinol (CBN)	0.000	0.000
Cannabidiolic Acid (CBDA)	14.464	144.640
Cannabidiol (CBD)	0.506	5.060
Delta-9-Tetrahydrocannabinol (Δ9THC)	0.057	0.570
Tetrahydrocannabinolic acid (THCA)	0.661	6.610
Total	16.157	161.570

**Table 2 life-10-00069-t002:** Cherry Blossom terpene analytics.

Phytochemical	% Total Weight	mg/g
Alpha-Humulene	0.036	0.360
Alpha-Pinene	0.048	0.480
Beta-Myrcene	0.256	2.560
Alpha-Bisabolol	0.040	0.400
Beta-Caryophyllene	0.121	1.210
Limonene	0.022	0.220
Guaiol	0.043	0.430
Farnesene	0.080	0.800
Total	0.646	6.460

Note. Only detectable terpenes at levels greater than 0.007% are shown above. Terpenes with non-detectable levels include: alpha-cedrene, alpha-terpinene, beta-pinene, borneol, camphene, camphor, caryophyllene oxide, cedrol, sabinene, sabinene hydrate, terpineol, terpinolene, trans-nerolidol, valencene, pulegone, alpha-phellandrene, ocimene, nerol, linalool, geranyl acetate, geraniol, gamma-terpinene, fenchone, eucalyptol, isoborneol, hexahydrothymol, fenchyl alcohol, 3-carene, cis-nerolidol, isopulegol.

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
