# Peer review of "The Therapeutic Effectiveness of Full Spectrum Hemp Oil Using a Chronic Neuropathic Pain Model"

_life, 2020, doi:10.3390/life10050069_

Round 1
Reviewer 1 Report
The necessity of therapeutic options to treat neuropathic pain is impelling and Hemp Oil option, which include low level of THC, could be usefull. Introduction well explain dark aspects of this field, however the section of methods and results is sparse. In discussion section, the authors argue well litterature data about anti-inflammatory and antiallodinic effects of single phytochemicals (CBD, terpenes such as β-myrcene, α-Pinene, and β-caryophyllene) and hypotize a mechanism of action of the hemp oil. However main compounds shown in table 1 are also CBDA (144.64 mg/g) and THCA (6.61 mg/g) but they are not mentioned in the discussion.
Main points:
- “The surgical model induction has been described in detail previously” in Reference 11 that refers to the Trigeminal Inflammatory Compression model. It could be usefull to elucidate difference between Foramen Rotundum Inflammatory Constriction Trigeminal Infraorbital Nerve injury and Trigeminal Inflammatory Compression models!! It is suggested to include a figure or scheme of the model used and, in case, point out the differences
- Reference 11 reports: “the small operating space and abundant blood supply in the facial area make the mouse infraorbital nerve injury model particularly challenging”: is the method used in the manuscript so available that is it enough and reproducible to use n=3 in Naive Control and FRICT-ION + PB groups?
- In method section is missed administration route of hemp. Correlate mg/g of the single phytochemical in the 5 microlitres volume.
- The caption of the figures and tables is missing; moreover it is usefull to report full name of the phytochemicals
- Timeline improuve: Referring to doi:10.3389/fphar.2017.00391 included in the references, insert acclimatization and Baseline Testing in timeline; lines 118-122 of the manuscript refer to 7th week but the asterisk is below 5. Point out dose, route of hemp administration and time.
- The manuscript could be grow rich comparing hemp oil effects with a dose-response curve of some of the main compounds present in the oil to verify the usefull of the full spectrum oil
Author Response
May 9, 2020
Re: Manuscript co-authored by JM Vigil, M Montera, NS Pentkowski, J Diviant, J Orozco, A Ortiz, L Rael, and KN Westlund, “The Therapeutic Effectiveness of Full Spectrum Hemp Oil Using a Chronic Neuropathic Pain Model”
Comments to author
Reviewer #1: The necessity of therapeutic options to treat neuropathic pain is impelling and Hemp Oil option, which include low level of THC, could be usefull. Introduction well explain dark aspects of this field, however the section of methods and results is sparse. In discussion section, the authors argue well litterature data about anti-inflammatory and antiallodinic effects of single phytochemicals (CBD, terpenes such as β-myrcene, α-Pinene, and β-caryophyllene) and hypotize a mechanism of action of the hemp oil. However main compounds shown in table 1 are also CBDA (144.64 mg/g) and THCA (6.61 mg/g) but they are not mentioned in the discussion.
à Thank you for the opportunity to expand details of our methods and results. We appreciate the Reviewer’s attention to the compounds, Cannabidiolic Acid (CBDA) and Tetrahydrocannabinolic acid (THCA), and we took the opportunity to briefly describe the limited literature on their potential role in pain attenuation on pg. 16: “Cannabidiolic acid (CBDA) and tetrahydrocannabinolic acid (THCA), the first and second most abundant cannabinoids (relatively) in the hemp plants used to produce the oil have not been extensively investigated for their therapeutic value, due in part to their molecular instability, but have both been shown to influence nociceptive and nonnociceptive mechanoreceptors.”
Main points:
- “The surgical model induction has been described in detail previously” in Reference 11 that refers to the Trigeminal Inflammatory Compression model. It could be usefull to elucidate difference between Foramen Rotundum Inflammatory Constriction Trigeminal Infraorbital Nerve injury and Trigeminal Inflammatory Compression models!! It is suggested to include a figure or scheme of the model used and, in case, point out the differences
à Thank you for the opportunity to provide further details on the distinction between the referenced and current procedures. The methods and results (New Figure 2) have been supplemented. A new FRICT-ION paper is now published and the reference provided in the text and reference list. Discussion in the Montera and Westlund paper points out the differences between several trigeminal neuropathic pain methods in the literature. A brief comparison appears in the Discussion of this re-written paper (pg. 14).
- Reference 11 reports: “the small operating space and abundant blood supply in the facial area make the mouse infraorbital nerve injury model particularly challenging”: is the method used in the manuscript so available that is it enough and reproducible to use n=3 in Naive Control and FRICT-ION + PB groups?
à Thank you also for suggesting that we elaborate on the reproducibility of our surgical procedure. Yes. The two controls never vary over time or during the test period. Also see the newly published protocol.
Montera, MA and Westlund KN. Minimally invasive oral surgery induction of the FRICT-ION chronic neuropathic pain model. Bio-protocol 2020, 10(08): e3591. doi:10.21769/BioProtoc.359
- In method section is missed administration route of hemp. Correlate mg/g of the single phytochemical in the 5 microlitres volume.
à The administration was oral ingestion in peanut butter. An additional method subsection is now dedicated to the oral dosing and controls (pgs. 7 and 8).
- The caption of the figures and tables is missing; moreover it is usefull to report full name of the phytochemicals
à It appears as though the caption and legends (and figures as per comments below) may have been misaligned in the version of the manuscript that the Reviewer’s viewed. There are captions for the figures and tables as appropriate, and we have ensured that the pdf copy of the manuscript displays these properly. We do appreciate the Reviewer’s suggestion to indicate both the full names and abbreviations of the cannabinoids in Table 1 and we now include both.
- Timeline improuve: Referring to doi:10.3389/fphar.2017.00391 included in the references, insert acclimatization and Baseline Testing in timeline; lines 118-122 of the manuscript refer to 7th week but the asterisk is below 5. Point out dose, route of hemp administration and time.
à Thank you for giving us the opportunity to provide further details on the project design. The Figure 1 Timeline has been edited to include the peanut butter and 6 hour testing
- The manuscript could be grow rich comparing hemp oil effects with a dose-response curve of some of the main compounds present in the oil to verify the usefull of the full spectrum oil
à Unfortunately extracting many of the individual compounds from the Cannabis plant remains difficult, hence the scarcity of such applications in the extant scientific literature (instead previous studies have mostly focused on synthetic analogues). The Reviewers comment is however very pertinent and we included this direction of future research in the limitations section on page 18: “Future research will benefit by comparing whole plant-extracted hemp oil effects with dose-response curves of some of the main compounds present in the oil to verify the utility of full spectrum therapeutics.”

Reviewer 2 Report
In this manuscript entitled "The therapeutic Effectiveness of Full Spectrum Hemp Oil Using a Chronic Neuropathic Pain Model" by Vigil et al. authors investigated the efficacy of the whole plant extract hemp oil in modulating mechanical allodynia in a model of chronic trigeminal neuropathic pain.
They found a potential restorative effect of hemp-extracted oil claiming that "the current findings offer tremendous promise for the layperson to be able to legally and relatively easily SELF-PRODUCE an ethnopharmaceutical product that is both effective and relatively safe".
First and foremost, the conclusions must be tempered given the observational nature of the work.
Second, the observation and limited scope of this manuscript. Reported evidences are basically behavioural analysis (mechanical threshold) on n = 12 animal subjects. I strongly suggest that discussion and conclusions must be tempered.
Third, authors assessed mechanical allodynia at 7 weeks post-surgery for 6 hrs. Why did they report just one timepoint? Did the authors study other timepoints? The time course of neuropathy is of major importance in evaluating chronic therapeutic regime.
Authors used a dose of 0.138 mg/kg. Please cite appropriate literature on this aspect.
I strongly suggest to re-build introduction and discussion also including novel frontiers in chronic pain management:
- Glutamatergic signalling (PMID: 27816787; PMID: 27585465; PMID: 30380983)
- Dopamine receptor antagonist (PMID: 18689859; PMID: 23815681)
- Serotonin neurotransmission (PMID: 25819610; PMID: 28690143)
- Opioid system (PMID: 31030416; PMID: 27590071; PMID: 21118955)
Experimental timeline – It is not aligned (maybe on my screen?); some boxes cover the text and I think that can be imporved.
Some minor points are related to spelling and on homogeneity throughout the manuscript (eg revise p value and use consistent way to report it 0.01 or .01).
Author Response
May 9, 2020
Re: Manuscript co-authored by JM Vigil, M Montera, NS Pentkowski, J Diviant, J Orozco, A Ortiz, L Rael, and KN Westlund, “The Therapeutic Effectiveness of Full Spectrum Hemp Oil Using a Chronic Neuropathic Pain Model”
Comments to author
Reviewer #2: First and foremost, the conclusions must be tempered given the observational nature of the work.
à We agree that the original language was overly conclusive and we have revised discussion of the interventions effectiveness in various locations of the manuscript, such as: “that common and commercially available full spectrum oil extracted from the hemp plant, using techniques that are generally replicable and accessible to the layperson, may be an effective treatment for chronic pain caused by mechanical pressure” on page 14.
Another example of tempered language on pg. 14 is here: “However, given that chronic pain remains a significant clinical challenge with a conventional treatment response rate of only 11%,34,35 the current findings show promising support for the effectiveness of a recently legalized (in the U.S.) alternative ethnopharmaceutical option for the treatment of some forms of chronic pain.”
IACUC committees and national agencies demand reduction in numbers of animals. Power analysis finds n=1 is sufficient to provide statistical significance with our model.
Second, the observation and limited scope of this manuscript. Reported evidences are basically behavioural analysis (mechanical threshold) on n = 12 animal subjects. I strongly suggest that discussion and conclusions must be tempered.
à As described above, we did take the opportunity to refine our interpretation of the results by more fully disclosing the limitation of the current small sample animal research design. We now include the Reviewers concern as an additional potential limitation of the study on page 18: “Finally, given the small sample and observational nature of the current research design, caution may be warranted when interpreting the generalizability of the findings and/or application to human conditions.”
Third, authors assessed mechanical allodynia at 7 weeks post-surgery for 6 hrs. Why did they report just one timepoint? Did the authors study other timepoints? The time course of neuropathy is of major importance in evaluating chronic therapeutic regime.
à Thank you for the opportunity to describe the current research design in further detail. The Figure 2 update provides the complete time course for this chronic neuropathic pain model.
Authors used a dose of 0.138 mg/kg. Please cite appropriate literature on this aspect.
à We appreciate the opportunity to provide further details on the Hemp dosing. We followed the manufacturers recommended dosing and calculated precisely how much that would be in mg/kg dosage (pgs. 7 and 8).
I strongly suggest to re-build introduction and discussion also including novel frontiers in chronic pain management:
- Glutamatergic signalling (PMID: 27816787; PMID: 27585465; PMID: 30380983)
- Dopamine receptor antagonist (PMID: 18689859; PMID: 23815681)
- Serotonin neurotransmission (PMID: 25819610; PMID: 28690143)
- Opioid system (PMID: 31030416; PMID: 27590071; PMID: 21118955)
à We appreciate the Reviewer’s suggestion to round out discussion of potential pharmacological mechanisms of action. On pg. 4 we describe: “Studies examining the neuropharmacology of neuropathic pain have implicated opioid (e.g., MOP/DOP),14–16 serotonin (e.g., 5-HT7),17,18 dopamine (e.g., D2)19,20 and glutamate (e.g., GluN2B)21–23 receptor systems as potential therapeutic targets, however, no studies to date have examined the effects of “full spectrum hemp oil on chronic pain.”
Experimental timeline – It is not aligned (maybe on my screen?); some boxes cover the text and I think that can be imporved.
à As per our observations noted above, I believe the view of the copy of ms the Reviewers saw was misformatted. This should now be fixed.
Some minor points are related to spelling and on homogeneity throughout the manuscript (eg revise p value and use consistent way to report it 0.01 or .01).
à Thank you, these minor points have been addressed.

Round 2
Reviewer 1 Report
The authors improuved thei manuscript
Reviewer 2 Report
Authors addressed my concerns, they also tempered conclusion and fully revised their manuscript.
I think that the manuscript is now suitable for publication.